# Influence of the Psychomotor Profile in the Improvement of Learning in Early Childhood Education

**DOI:** 10.3390/ijerph182312655

**Published:** 2021-11-30

**Authors:** Francisco José Borrego-Balsalobre, Alfonso Martínez-Moreno, Vicente Morales-Baños, Arturo Díaz-Suárez

**Affiliations:** Department of Physical Activity and Sport, CEI Campus Mare Nostrum, University of Murcia, 30100 Murcia, Spain; franborrego@um.es (F.J.B.-B.); ardiaz@um.es (A.D.-S.)

**Keywords:** psychomotor profile, evaluation, child education, academic performance

## Abstract

The development of psychomotor skills in childhood enables children to organise the outside world through their bodies, contributing to their intellectual, affective, and social development. The present study aimed to longitudinally evaluate the psychomotor profile, throughout three academic years, of 3, 4 and 5-year-olds belonging to the second cycle of infant school, relating it descriptively to academic performance. The sample consisted of 82 subjects aged between 3 and 6 years throughout the study. The distribution of the sample was homogeneous, with 47.6% boys (n = 39) and 52.4% girls (n = 43). The results not only highlight the importance of the development and stimulation of motor skills from an early age for the overall development of the child, but also, when related to previous studies, show how they influence the development of human beings in adulthood.

## 1. Introduction

Scientific literature establishes that early childhood education lays the foundations of children’s personal and social development, as it is at this stage that the learning that serves as a foundation for the achievement of competencies considered basic for the development of the person is integrated [1,2,3]. Competence is understood as the child’s ability to put into practice, in an integrated way, in different contexts and situations, both theoretical knowledge and practical skills or knowledge, as well as acquired personal attitudes. The concept of competence encompasses more than just knowing how to do or how to apply, as it also includes knowing how to be [4]. Basic competences are, therefore, the knowledge, skills, and attitudes that all individuals need, both for their personal fulfilment and development, as well as for their inclusion in society [5].

In this sense, by their very nature, they are closely linked to the course of life and acquired experience and cannot be required at an initial level. On the contrary, it is necessary to promote their development by working on them from an early age [6]. This reasoning acquires its maximum expression insofar as reference is made to the addition of this concept in the definition of the educational curriculum. Therefore, a detailed study on how the basic competences are to be worked on in the child’s daily practice is essential, in view of the fact that the acquisition and improvement of these competences throughout the different educational stages is of crucial importance. The school serves as an instrument for monitoring and enhancing the degree of attainment of these competences. To this end, these competences should be acquired by the end of compulsory education and should form the basis of continuous lifelong learning [7].

Following this approach, at the stage of infant education, psychomotor skills play a fundamental role, insofar as they help children to mentally organise the outside world through their bodies. Their intellectual, affective, and social development is influenced by facilitating their relationship with the environment in different settings, contexts, and situations. Numerous authors confirm that, at an early age, anatomical-physiological and affective-intellectual structures are developed together, as they are associated in such a way that they really constitute a single unit. Psychomotor skills and their early development help children to master their body through balance and movement, preparing them for the motor needs of the environment and daily life. On the other hand, at a cognitive level, it allows them to improve their attention span and concentration, as well as to memorise and encourage the development of their creativity. Likewise, on a social level, it allows them to interact with others more frequently, helping them to go out of their usual environment [8,9,10,11].

Thus, if the aim is to achieve a good all-round education of the child, it is of vital importance to consider the addition of an appropriate psychomotor intervention programme from an early age to the teaching programme [12]. To this end, prior to the start of any type of educational programme, it is necessary to evaluate the level reached by the pupils in the aspects included in it [13,14]. This is particularly important in the case of psychomotor education, considering that the performance of a behaviour in this area is not achieved before having passed the one that precedes it (standing and walking, jumping, etc.) [15].

In this regard, from the many assessment instruments available in the scientific literature for the evaluation of the psychomotor profile in children, the following five areas or variables are the most frequently mentioned by the authors: locomotion, positions, balance, coordination, and knowledge of the body schema [16,17,18,19,20,21]. To that effect, locomotion can be understood as the different ways in which the child moves or changes position by their own means. On the other hand, posture is understood as the ability to adopt and maintain a certain position in a standing or walking position. Balance is defined as the ability to hold a position in the least possible contact with the surface. Coordination, whether of legs, arms, or hands, is understood as the simultaneous use of several muscles (fine motor skills) or groups of muscles (gross motor skills). Finally, the body scheme assesses the knowledge of the body seen in oneself or in others [20].

There are also numerous studies that link adequate psychomotor development through early stimulation with academic achievement and desirable classroom behaviour in preschool and later childhood. Thus, active play during recess is associated with self-regulation and academic achievement [22,23]. At school, many of the cognitive processes observed in pupils that directly influence academic performance are: self-concept, self-esteem, extrinsic and intrinsic motivation, and prosocial attitudes. Likewise, regular physical activity and structured physical education allow for the development of children’s motor and mental skills during early childhood, proving to be of vital importance for the psychomotor development of pre-school children, and verifying its relevance for children’s relationships with the outside world [24,25]. This is fundamental for children’s knowledge of themselves and their environment. Similarly, some studies conclude that there are benefits in mathematics and reading because of physical activity in children [14,26,27].

Therefore, it is important to work on all these processes through the area of physical education, so that children grow up with a positive perception of themselves. Successful experiences influence the effectiveness of doing things, so self-confidence will prepare them to tackle new challenges. It is therefore necessary to consider how psychomotor development and the acquisition of academic skills together influence the integral development of the child [13,28,29,30,31].

From the theoretical foundation carried out, the need arises to elaborate specific proposals that relate the different motor skills based on the knowledge of the body scheme, with the cognitive skills acquired by children from an early age, insofar as these proposals tend to be generic, not establishing relationships of the different parameters that make up the psychomotor profile with the corresponding cognitive abilities. For that reason, the aims of the present work are:

To evaluate the psychomotor profile longitudinally across three academic years, ensuring that the sample under study passes through the 3-, 4- and 5-year-old infant school years.

To find out how basic competences evolve within the same academic year and throughout the three years by means of both the analysis of motor competences and academic performance through the following three constructs: self-awareness, language and communication and knowledge of the environment.

To find out the relationship that can be established between the different dimensions that make up the psychomotor profile of the sample under study.

To find out the relationship that can be established between the psychomotor profile and the academic performance of the sample under study.

## 2. Materials and Methods

### 2.1. Design and Participants

This was a descriptive, comparative, and longitudinal study. The final sample that completed all the measurements carried out was composed of 82 subjects, belonging to the 3-year-olds in the first academic year of the study (2018–19), the 4-year-olds in the second year (2019–20) and the 5-year-olds in the third year (2020–21); all of them being boys and girls who completed the second cycle of infant school in the three years, aged between 3 and 6 years throughout the study. The distribution of the sample was homogeneous, with 47.6% boys (n = 39) and 52.4% girls (n = 43).

Participants were selected by convenience because, to carry out the study, the school with the highest number of students enrolled in preschool in the entire region was intentionally selected. The option of participating in the study in the first year was proposed to all pupils in the 3-year-old infant school year belonging to the five lines or groups to which the study had access, establishing as inclusion-exclusion criteria that they should bring a duly completed informed consent form and that they should not have medical indications that would prevent them from taking part in a normal physical education session.

The data collected from participants who did not complete the study were discarded, since, over the course of the three academic years that the study lasted, there were students who stopped participating in the study due to various circumstances. This process is summarized in Figure 1.

### 2.2. Variables and Instruments

Assessment of the Psychomotor Profile. For this variable, the Psychomotor Assessment Scale for Pre-schoolers [20] was used to assess the following aspects of psychomotor development, which made it possible to measure a total of 40 items: locomotion (L); positions (P); balance (B); leg coordination (LC); arm coordination (AC); hand coordination (HC); body schema self-awareness (BSA); and body schema other-awareness (BSoA). Understood as low (does not do it), normal (does it sometimes), and good (does it always). Likewise, started (does it halfway) in progress (does it sometimes), and achieved (does it always).

Assessment of academic performance. Through the assessment bulletins handed in by the teachers in the second and third respective trimester of each academic year, in which the following areas are assessed: self-awareness (SA), language (LG), and knowledge of the environment (KE).

### 2.3. Procedure

Several prior meetings were held with the management and teaching staff involved, in order to organize the data collection process. Likewise, different meetings of experts were held with the researchers in charge of data collection, with the aim of refining the data collection, training them, and being able to unify measurement criteria based on the protocol and application rules established for this purpose. At these meetings, the measurement protocol was carefully reviewed, as were the sections on instructions for the evaluator as well as the necessary materials for the measurement. In parallel, measurement tests were carried out among the evaluators to ensure the same interpretation for each of the items to be evaluated by all the researchers and to debug any possible setbacks that might arise.

Prior to the start of the measurements, a meeting was held with the parents and/or guardians who had been provided with initial information by letter and who had decided to participate in the study to clarify all the questions they wished to make.

During the three academic years in which the study was carried out, data were always collected twice, two months apart. The first measurement was taken in February-March and the second in April-May. To this end, in previous meetings with the teaching team involved, dates and times were set for the assessments to be carried out during school hours, always trying to respect two premises: firstly, the dates proposed by the centre and, secondly, the grouping of pupils by year group. Thus, an assessment calendar was established for each year.

### 2.4. Data Analysis

Qualitative variables were described by absolute (n) and relative (%) frequencies; mean and standard deviation (SD) for quantitative variables. For the study of the evolution of children’s psychomotor skills according to gender, a two-factor ANOVA test with repeated measures in one of them was carried out using the General Linear Model (GLM) procedure. The dimensions of performance were studied using the Wilcoxon test. The logistic regression model was used to determine the effect of the improvement in the dimensions of psychomotor skills on the improvement in each of the performance dimensions: SA, LG, and KE. Statistical analysis was performed with SPSS 25.0 for Windows. The differences considered statistically significant are those whose *p* < 0.05.

### 2.5. Ethical Aspects

All participating subjects (through their parents and/or guardians) signed an informed consent form indicating that the data collected would be processed anonymously, and the corresponding authorisation was requested from the Bioethics Committee of the University of Murcia in accordance with the ethical principles reflected in the different treaties and official documents to guarantee the strict confidentiality and professional ethics of educational research [32]. Even so, the University’s Research Ethics Committee considered that a favourable report from the Committee was not mandatory, as the data in question were not likely to infringe the fundamental rights of the subjects under study. The Ethics Commission determined that, as it was an observational study and reviewed the tests performed, it was not necessary for the commission to issue a report.

## 3. Results

Table 1 shows the mean attained pretest and posttest, as well as time in relation to gender and the within-subjects effect of time. The time effect shows statistically significant differences between pretest and posttest in the 3-year-old subjects, in B and LC, and L also shows these differences when comparing time in relation to gender, with girls achieving higher scores in the posttest. In the 4-year-olds, statistically significant differences are found in L, P, B, LC, AC, HC, and BSA in terms of time. The rest of the dimensions do not indicate statistically significant differences, as well as when compared to gender.

In 5-year-olds, the time effect shows statistically significant differences in the dimensions L, P, B, LC, and BSoA. The rest of the dimensions do not indicate statistically significant differences, as well as when compared to gender.

Table 2 shows the evaluation of the level of psychomotor skills in each of the academic year under study. In relation to the 3-year-olds, in the dimensions of L, P, B, and LC, the level is mostly good, being normal in AC, HC, BSA, and BSoA. As for the low score, very few 3-year-olds are at this level. In 4 years, the level of good appears predominantly in L, P, B, and BSA, reaching the normal level in the dimensions LC, AC, HC, and BSoA. The score of low is very low in all dimensions, except in BSA where 18.3% of subjects obtain this score. As for 5-year-old pupils, in the good level are the dimensions L, B, BSA, and BSoA, and in the normal level we find the dimensions P, LC, AC, HC, and BSA. When we analyse the low level, we see that it appears in a testimonial way in all the dimensions except in AC, where there are 31.7% of subjects, although it is not the majority score.

The absolute and relative frequencies in the pre-test and post-test, Wilcoxon test in relation to the academic variables, Table 3, are detailed next. All the dimensions in relation to the pre-test and post-test in pupils aged 3, 4 and 5 present statistically significant differences. In relation to the pre-test in 3 years, most of the students are in the “initiated” option in the three academic dimensions SA, LG, and KE, with most of the students in the post-test in 3 years indicating the “in progress” option. As for the 4-year-olds in the pre-test, almost half of the pupils are “in progress” in relation to the dimensions analysed, and in the post-test half of the pupils are “in progress” and a quarter have “achieved” the expected indicators. When analysing the 4-year-olds, one third have already “achieved” the objectives in the pre-test and in the post-test 100% of the pupils achieve the objectives of the dimensions SA, LG, and KE.

To find out the effect that each of the psychomotor dimensions analysed (L, P, B, LC, AC, HC, BSA, and BSoA) may have on the academic variables (SA, LG, and KE), a regression was performed (Table 4) for pupils aged 3, 4, and 5 years. As for 3-year-olds, the improvement in SA has a statistically significant effect on HC. Thus, the children in the sample who improve in HC are 15.66 (OR = 15.66, *p* = 0.02) times more likely to improve in SA than those who do not improve in HC. In the improvement in LG, the dimensions of B, LC, HC, and BSoA have a statistically significant effect. Thus, children who improve in B are less likely to improve in language (OR = 0.14, *p* = 0.018). On the other hand, those who improve in LC are 7.71 (OR = 7.71, *p* = 0.014) times more likely to improve in LG than those who do not improve in LC. Similarly, those who improve in HC are 36.4 times more likely to improve in LG than those who do not improve in HC (OR = 36.4, *p* = 0.005). Finally, those who improve in BSoA are 20.4 times more likely to improve in LG than those who do not improve in BSoA (OR = 20.4, *p* = 0.018). As for improvement in KE, it has a statistically significant effect on all dimensions of psychomotor skills. Thus, children who improve in L (OR = 0.26, *p* < 0.001), B (OR = 0.60, *p* < 0.001), and AC (OR = 0.79, *p* < 0.001) are less likely to improve in KE. While those who improve in P are 1.43 times, those who improve in LC are 2.59 times, those who improve in HC are 8.89 times, those who improve in BSA are 2.97 times, and those who improve in BSoA are 1.69 times more likely to improve in KE than those who do not improve in each of the above.

When analysing the results of the 4-year-olds in relation to SA, there is a statistically significant effect in AC and BSA, in terms of LG the statistically significant effect is in relation to P, AC, and BSA, and in terms of KE all the variables have a statistically significant effect. In such a way that those who improve in AC and BSA are between 5.90 and 11.75 times more likely to improve in SA, LG, and KE than those who do not improve in AC and BSA respectively. In addition, those who improve in P are 3.07 times more likely to improve in LG and 2.85 times more likely to improve in KE than those who do not improve in P. Similarly, those who improve in L, B, HC, and BSoA are between 1.80 and 6.67 times more likely to improve in KE than those who do not improve in L (OR = 2.80, *p* < 0.001); B (OR = 6.67, *p* < 0.001); HC (OR = 1.80, *p* < 0.001) and BSoA (OR = 1.93, *p* < 0.001).

When comparing the data in 5-year-olds, improvement in SA has a significant effect on B and BSoA. Thus, those who improve in B are 16.41 (OR = 16.41, *p* < 0.001) times more likely and those who improve in BSoA are 4.76 (OR = 4.76, *p* = 0.037) times more likely to improve in SA than those who do not improve in B and BSoA. In relation to LG, those who improve in B are 7.52 times more likely to improve in LG than those who do not improve in B (OR = 7.52, *p* = 0.002). Regarding KE, those who improve in L, P, B, LC, AC, HC, BSA, and BSoA are between 1.29 and 7.88 times more likely to improve in KE than those who do not improve in the variables.

## 4. Discussion

The present study evaluated the cross-sectional psychomotor profile of infant children during their passage from 3 to 5 years of age, who showed significant improvements at the 3-year motor development stage only in balance and coordination of the lower body because of their own growth, with girls showing significant improvements also in locomotion. This is believed to be a consequence of the fact that in very young children, immature control of posture and gait leads to unstable locomotion, and it is not until the age of three that gait begins to become relatively mature, although it is not known whether the dynamics of walking change beyond this age [33]. In this sense, at 3 years of age, the child has already acquired many of the motor skills of an adult such as running, jumping, and climbing. However, some of the skills acquired in earlier stages are still being developed and perfected [34].

From this stage onwards, the knowledge of the world around them begins to be structured, generating schemes of the most common situations for them, thus achieving a broader knowledge of their environment. There is a progressive improvement in gross motor skills and a special development of fine motor skills. In this sense, the subjects evaluated at the 4-year motor development stage showed significant improvements, not only in balance and lower body coordination, but also in locomotion, positions, balance, hand coordination, and body schema, in accordance with the gradual progression of the motor development milestones [35].

This tendency to improve the parameters of the psychomotor profile because of maturation itself continues to be shown by the subjects under study at the 5-year stage, generating maturational effects in locomotion, positions, balance, and coordination in the lower body, but also, from this stage onwards, the body schema in others also becomes important. The children, in their evolutionary process, develop the correct structuring of their body schema. The child up to the age of 3 is characterised by the discovery of the child’s own body and refining their perception during the infant stage, so that once they reaches primary school, they can correctly represent their own body in movement, becoming aware of the dynamism it acquires [36]. Hence, in the final stages of this process, they can transfer what they have been identifying in their own body to the body of others, coinciding with the stage known as the definitive elaboration of the body schema or represented body [37].

Following the above, it is therefore straightforward that in the results obtained from the subjects of the present study, in relation to the academic variables, in the pre-test of the 3-year stage, in the dimension of knowledge of oneself and one’s environment, the value initiated predominates, reaching, in the post-test, the value “in progress” for the most part. Likewise, it is also well founded the fact that at 4 years of age, in the pre-test, the predominant value is the “in progress” value, which increases in the post-test. However, the value of “achieved” is not predominant yet, which does occur in the 5-year-old stage.

The analysis of the data also shows the relationship between the development of hand coordination and the academic variable of self-awareness, especially at the beginning of the infant stage, insofar as the infant mastery of the hands requires a high level of precision, training the children to be able to achieve other more complex hand skills. In this sense, as children grow, they acquire the set of skills that make up fine psychomotor skills either naturally or with help, which will allow them to acquire greater knowledge and motivation towards what they can achieve [38].

Likewise, in the present study, it can be observed that the improvement of fine motor skills through hand coordination is related to the improvement of the academic variable “language”, in view of the fact that for children to learn to pick up a pencil and begin to trace and write, they must first have developed their ability to handle their hands and fingers well, as well as the synchronisation of their movements [39]. Adequate hand control allows for the proper development of reading and writing. The child must work with different materials to achieve the appropriate level of precision and coordination required for general tasks, especially those involving simultaneous use of eyes, hands and fingers. In the same way, it is necessary to carry out exercises that encourage the development of hand-eye coordination that will lead the child to mastery of the hand and the other elements involved in its movement (wrist, forearm, and arm), leveraging the development and enhancement of motivational aspects through their achievement orientation [40].

However, it is also important to point out that the tendency shown by the data referring to the improvement of the academic variable knowledge of the environment is attributed to children who achieve higher scores in the improvement of practically all the parameters that make up the psychomotor profile. Thus, obtaining improvements in this profile will allow the child to be in greater contact with the environment. The motor development of children will depend above all on the global maturation of the body and the skeletal and neuromuscular development, resulting in greater control of the body and the environment because of the achievements they acquire, in turn influencing both their ability to relate and to express themselves [41]. To this end, play and spontaneous movement play a fundamental role in the child’s development.

All these results and previous studies show the importance of the development and stimulation of children’s motor skills at an early age. Thus, many authors have been investigating the effects of physical activity on cognitive skills, highlighting the benefits for higher cognitive functions. Among these findings we encounter that factors such as attention, language, memory, processing speed, perception, and thinking develop more easily [36,41,42]. Thus, being able to carry out a longitudinal proposal such as the one in the present study in which, by establishing control groups, different intervention programmes can be carried out between the pre-test and post-test throughout the three pre-school child stages could contribute to finding out to what extent children who exercise have a better capacity to regulate their general cognitive skills, greater reaction capacity and a better level of attention to discriminate relevant stimuli from those that do not, while also analysing whether good psychomotor development and stimulation can be a good predictor of more complex learning skills later in life.

## 5. Conclusions

Motor activity and movement adapt human beings to reality, playing a fundamental role in both affective and social life from an early age. It is in the infant stage that most of the changes that allow children to explore the world around them take place. This interaction in space-time is acquired through adequate psychomotor development that allows the child to respond to the different challenges presented to them. However, there are different parameters that make up the child’s psychomotor profile, which must be matured and acquired at the corresponding and appropriate moments in the process. Promoting these through physical activity, favouring the child’s development, gives them the opportunity to develop their skills, which will allow them to solve more complex situations in later stages, providing autonomy and benefits not only on an emotional level but also on an academic level. Thus, motor development contributes to learning structures and maturation schemes where the child’s functions can reach certain habits, skills, knowledge in matrix operations and abilities that will be of vital importance for their daily life.

The main limitation of the study, which could have provided another vision to it, was not carrying out a sport’s physical activity program during the two months that elapsed between the pre-test and the post-tests in the same academic year, since this could allow establish cause-effect relationships year by year and throughout the entire study in relation to the evolution of the motor patterns of preschool-age children.

## Figures and Tables

**Figure 1 ijerph-18-12655-f001:**
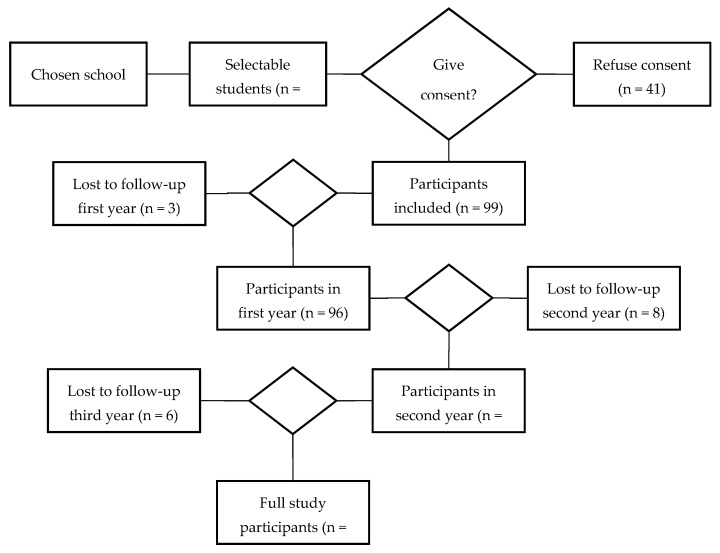
Flow chart of sample lost along the study.

**Table 1 ijerph-18-12655-t001:** Means (SD) and statistical contrasts between genders in the psychomotor skills scale in 3-, 4-, and 5-year-old pupils.

	3-Year-Olds				4-Year-Olds				5-Year-Olds			
	Measure, Means *(SD)*		Within-Subjects Effect		Measure, Means *(SD)*		Within-Subjects Effect		Measure, Means *(SD)*		Within-Subjects Effect	
	PRE	POST	Time	Gender-Time	PRE	POST	Time	Gender-Time	PRE	POST	Time	Gender-Time
			*F*(d.f.); *p*-Value (η^2^)	*F*(d.f.); *p*-Value (η^2^)			*F*(d.f.); *p*-Value (η^2^)	*F*(d.f.); *p*-Value (η^2^)			*F*(d.f.); *p*-Value (η^2^)	*F*(d.f.); *p*-Value (η^2^)
**Locomotion**			F (1;80) = 3.72; *p* = 0.057 (0.044)	F (1;80) = 6.09; *p* = 0.016 (0.071)			F (1;80) = 18.52; *p* < 0.001 (0.188)	F (1;80) = 0.04; *p* = 0.834 (0.001)			F (1;80) = 26.86; *p* < 0.001 (0.251)	F (1;80) = 0.13; *p* = 0.721 (0.002)
Male	12.56 (1.3)	12.46 (2.1)			12.87 (1.2)	13.67 (0.6)			13.15 (1.0)	13.79 (0.5)		
Female	12.05 (1.8)	12.88 (1.3)			12.79 (1.7)	13.51 (0.9)			13.47 (0.9)	14.02 (0.8)		
*Total*	12.29 (1.6)	12.68 (1.7)			12.83 (1.5)	13.59 (0.8)			13.32 (0.9)	13.91 (0.7)		
**Positions**			F (1;80) = 3.21; *p* = 0.077 (0.039)	F (1;80) = 1.60; *p* = 0.210 (0.020)			F (1;80) = 12.36; *p* = 0.001 (0.134)	F (1;80) = 2.64; *p* = 0.108 (0.032)			F (1;80) = 32.11; *p* < 0.001 (0.286)	F (1;80) = 0.54; *p* = 0.466 (0.007)
Male	4.82 (1.5)	5.36 (1.0)			4.95 (0.8)	5.21 (1.2)			5.44 (0.7)	5.95 (0.3)		
Female	5.12 (1.0)	5.21 (1.1)			4.86 (0.9)	5.56 (0.7)			5.58 (0.7)	5.98 (0.2)		
*Total*	4.98 (1.3)	5.28 (1.1)			4.90 (0.8)	5.39 (1.0)			5.51 (0.7)	5.96 (0.2)		
**Balance**			F (1;80) = 53.91; *p* < 0.001 (0.403)	F (1;80) = 2.65; *p* = 0.107 (0.032)			F (1;80) = 13.14; *p* = 0.001 (0.141)	F (1;80) = 0.07; *p* = 0.792 (0.001)			F (1;80) = 66.08; *p* < 0.001 (0.452)	F (1;80) = 1.10; *p* = 0.298 (0.014)
Male	9.33 (1.6)	10.67 (2.0)			8.77 (2.1)	9.85 (2.1)			9.74 (1.4)	11.46 (0.7)		
Female	8.72 (2.0)	10,.81 (1.7)			8.86 (1.8)	9.79 (2.6)			10.23 (1.3)	11.56 (1.1)		
*Total*	9.01 (1.8)	10.74 (1.9)			8.82 (1.9)	9.82 (2.4)			10.00 (1.4)	11.51 (0.9)		
**Leg Coord.**			F (1;80) = 13.85; *p* < 0.001 (0.148)	F (1;80) = 2.14; *p* = 0.148 (0.026)			F (1;80) = 65.27; *p* < 0.001 (0.449)	F (1;80) = 0.64; *p* = 0.427 (.008)			F (1;80) = 24.80; *p* < 0.001 (0.237)	F (1;80) = 2.57; *p* = 0.113 (0.031)
Male	10.77 (1.3)	11.21 (1.3)			11.67 (1.0)	9.26 (2.3)			11.33 (1.0)	11.92 (0.4)		
Female	10.44 (1.9)	11.44 (1.2)			11.49 (1.3)	9.51 (2.3)			11.70 (0.5)	12.00 (0.0)		
*Total*	10.60 (1.6)	11.33 (1.2)			11.57 (1.2)	9.39 (2.3)			11.52 (0.8)	11.96 (0.2)		
**Arm Coord.**			F (1;80) = 1.26; *p* = 0.265 (0.015)	F (1;80) = 2.24; *p* = 0.138 (0.027)			F (1;80) = 11.63; *p* = 0.001 (0.127)	F (1;80) = 1.16; *p* = 0.286 (0.014)			F (1;80) = 0.30; *p* = 0.583 (0.004)	F (1;80) = 0.40; *p* = 0.529 (0.005)
Male	7.13 (1.5)	6.64 (1.7)			6.28 (1.9)	7.67 (2.3)			8.59 (1.7)	8.56 (1.9)		
Female	6.49 (1.8)	6.56 (1.4)			6.14 (1.8)	6.86 (2.0)			8.53 (1.6)	8.91 (1.7)		
*Total*	6.79 (1.7)	6.60 (1.5)			6.21 (1.9)	7.24 (2.2)			8.56 (1.6)	8.74 (1.8)		
**Hand Coord.**			F (1;80) = 0.43; *p* = 0.513 (0.005)	F (1;80) = 0.13; *p* = 0.725 (0.002)			F (1;80) = 17.51; *p* < 0.001 (0.180)	F (1;80) = 0.01; *p* = 0.936 (0.000)			F (1;80) = 2.11; *p* = 0.151 (0.026)	F (1;80) = 0.23; *p* = 0.631 (0.003)
Male	7.00 (2.1)	6.92 (1.9)			8.67 (1.6)	9.51 (1.0)			9.49 (0.8)	9.62 (0.6)		
Female	7.28 (2.0)	7.02 (2.2)			8.65 (1.6)	9.47 (1.0)			9.26 (1.1)	9.51 (0.7)		
*Total*	7.15 (2.1)	6.98 (2.1)			8.66 (1.6)	9.49 (1.0)			9.37 (1.0)	9.56 (0.6)		
**BSA**			F (1;80) = 3.13; *p* = 0.081 (0.038)	F (1;80) = 0.11; *p* = 0.737 (0.001)			F (1;80) = 21.53; *p* < 0.001 (0.212)	F (1;80) = 0.15; *p* = 0.705 (0.002)			F (1;80) = 2.74; *p* = 0.102 (0.033)	F (1;80) = 2.02; *p* = 0.159 (0.025)
Male	6.92 (2.6)	7.54 (1.6)			4.59 (3.1)	6.97 (3.4)			8.72 (2.2)	8.79 (2.5)		
Female	6.93 (2.5)	7.35 (1.5)			5.07 (2.8)	7.09 (3.3)			8.09 (2.8)	9.12 (2.0)		
*Total*	6.93 (2.5)	7.44 (.5)			4.84 (2.9)	7.04 (3.3)			8.39 (2.5)	8.96 (2.2)		
**BSoA**			F (1;79) = 0.89; *p* = 0.349 (0.011)	F (1;79) = 3.59; *p* = 0.062 (0.043)			F (1;80) = 1.62; *p* = 0.206 (0.020)	F (1;80) = 1.35; *p* = 0.249 (0.017)			F (1;80) = 143.65; *p* < 0.001(0.642)	F (1;80) = 0.60; *p* = 0.441 (0.007)
Male	3.84 (1.9)	3.63 (2.1)			3.90 (1.4)	3.87 (1.8)			3.36 (1.6)	5.82 (0.6)		
Female	3.23 (1.5)	3.86 (1.9)			4.16 (1.6)	3.60 (1.7)			3.58 (1.6)	5.74 (0.9)		
*Total*	3.52 (1.7)	3.75 (2.0)			4.04 (1.5)	3.73 (1.7)			3.48 (1.6)	5.78 (0.7)		

d.f.: degrees of freedom. η^2^: partial eta square.

**Table 2 ijerph-18-12655-t002:** Psychomotor assessment at 3, 4, and 5 years of age.

	Psychomotor Skills Level, n (%)								
	3 Years			4 Years			5 Years		
	Good	Normal	Low	Good	Normal	Low	Good	Normal	Low
Locomotion	52 (63.4)	29 (35.4)	1 (1.2)	74 (90.2)	8 (9.8)	0 (0)	72 (87.8)	10 (12.2)	0 (0)
Positions	49 (59.8)	30 (36.6)	3 (3.7)	49 (59.8)	31 (37.8)	2 (2.4)	0 (0)	81 (98.8)	1 (1.2)
Balance	57 (69.5)	22 (26.8)	3 (3.7)	47 (57.3)	29 (35.4)	6 (7.3)	47 (57.3)	35 (42.7)	0 (0)
Leg Coord.	55 (67.1)	27 (32.9)	0 (0)	24 (29.3)	54 (65.9)	4 (4.9)	0 (0)	81 (98.8)	1 (1.2)
Arm Coord.	4 (4.9)	75 (91.5)	3 (3.7)	19 (23.2)	60 (73.2)	3 (3.7)	0 (0)	56 (68.3)	26 (31.7)
Hand Coord.	8 (9.8)	66 (80.5)	8 (9.8)	54 (65.9)	27 (32.9)	1 (1.2)	0 (0)	82 (100)	0 (0)
BSA	7 (8.5)	74 (90.2)	1 (1.2)	42 (51.2)	25 (30.5)	15 (18.3)	52 (63.4)	25 (30.5)	5 (6.1)
BSoA	25 (30.5)	46 (56.1)	11 (13.4)	28 (34.1)	53 (64.6)	1 (1.2)	74 (90.2)	5 (6.1)	3 (3.7)

**Table 3 ijerph-18-12655-t003:** Absolute and relative frequencies, Wilcoxon test in relation to academic variables.

	PRE			POST			Wilcoxon Test	
	Initiated	In Progress	Achieved	Initiated	In Progress	Achieved	Z	*p*-Value
**3** **years**								
SA	69 (84.1)	13 (15.9)	0 (0)	8 (9.8)	56 (68.3)	18 (22)	−7.47	<0.001
LG	68 (82.9)	9 (11)	5 (6.1)	8 (9.8)	49 (59.8)	25 (30.5)	−7.00	<0.001
KE	64 (78)	10 (12.2)	8 (9.8)	8 (9.8)	45 (54.9)	29 (35.4)	−6.87	<0.001
**4** **years**								
SA	34 (42)	47 (58)	0 (0)	2 (2.4)	54 (65.9)	26 (31.7)	−6.32	<0.001
LG	36 (44.4)	45 (55.6)	0 (0)	3 (3.7)	51 (62.2)	28 (34.1)	−6.45	<0.001
KE	36 (44.4)	45 (55.6)	0 (0)	2 (2.4)	55 (67.1)	25 (30.5)	−6.51	<0.001
**5** **years**								
SA	8 (9.8)	41 (50)	33 (40.2)	0 (0)	0 (0)	82 (100)	−6.58	<0.001
LG	10 (12.2)	38 (46.3)	34 (41.5)	0 (0)	0 (0)	82 (100)	−6.44	<0.001
KE	8 (9.8)	38 (46.3)	36 (43.9)	0 (0)	0 (0)	82 (100)	−6.36	<0.001

**Table 4 ijerph-18-12655-t004:** Effect of the improvement of psychomotor dimensions on the improvement of academic dimensions at 3, 4, and 5 years old.

**3 Years**						
	**Self-Awareness**		**Language**		**Knowledge of the** **Environment**	
	**OR (IC 95%)**	***p*-Valor**	**OR (IC 95%)**	***p*-Valor**	**OR (IC 95%)**	***p*-Valor**
Locomotion	0.35 (0.08–1.64)	0.185	0.62 (0.12–3.23)	0.569	0.26 (0.06–1.23)	<0.001
Positions	1.48 (0.37–6.00)	0.583	0.51 (0.10–2.47)	0.399	1.43 (0.33–6.22)	<0.001
Balance	0.26 (0.06–1.12)	0.071	0.14 (0.03–0.71)	0.018	0.60 (0.16–2.29)	<0.001
Leg Coord.	1.30 (0.35–4.84)	0.701	7.71 (1.50–39.66)	0.014	2.59 (0.66–10.10)	<0.001
Arm Coord.	1.41 (0.29–6.82)	0.668	0.96 (0.16–5.85)	0.966	0.79 (0.18–3.40)	<0.001
Hand Coord.	15.66 (1.53–160.11)	0.02	36.40 (2.99–443.48)	0.005	8.89 (1.27–62.43)	<0.001
BSA	1.02 (0.25–4.25)	0.975	0.65 (0.13–3.25)	0.596	2.97 (0.62–14.17)	<0.001
BSoA	4.90 (0.86–28.06)	0.074	20.40 (1.70–245.55)	0.018	1.69 (0.36–7.89)	<0.001
R^2^ of Nagelkerke	0.284		0.458		0.224	
Model	*χ*2(8) = 16.42; *p* = 0.037		*χ*2(8) = 29.06; *p* < 0.001		*χ*2(8) = 12.67; *p* = 0.124	
**4 Years**						
	**Self-Awareness**		**Language**		**Knowledge of the** **Environment**	
	**OR (IC 95%)**	***p*-Valor**	**OR (IC 95%)**	***p*-Valor**	**OR (IC 95%)**	***p*-Valor**
Locomotion	2.88 (0.72–11.52)	0.135	1.55 (0.42–5.76)	0.51	2.80 (0.73–10.72)	<0.001
Positions	3.18 (0.93–10.92)	0.066	3.07 (0.96–9.87)	0.059	2.85 (0.86–9.43)	<0.001
Balance	2.77 (0.79–9.71)	0.112	1.98 (0.61–6.43)	0.258	6.67 (1.81–24.54)	<0.001
Leg Coord.	11.75 (3.45–40.06)	<0.001	8.89 (2.64–29.97)	<0.001	5.72 (1.76–18.57)	<0.001
Arm Coord.	1.16 (0.33–4.06)	0.819	0.82 (0.24–2.78)	0.751	1.80 (0.52–6.23)	<0.001
Hand Coord.	5.90 (1.57–22.15)	0.009	6.78 (1.95–23.62)	0.003	6.46 (1.78–23.55)	<0.001
BSA	1.45 (0.44–4.73)	0.543	1.50 (0.47–4.76)	0.496	1.93 (0.59–6.29)	<0.001
BSoA	2.88 (0.72–11.52)	0.135	1.55 (0.42–5.76)	0.51	2.80 (0.73–10.72)	<0.001
R^2^ of Nagelkerke	0.492		0.44		0.458	
Model	*χ*2(7) = 37.13; *p* < 0.001		*χ*2(7) = 32.18; *p* < 0.001		*χ*2(7) = 33.79; *p* < 0.001	
**5 Years**						
	**Self-Awareness**		**Language**		**Knowledge of the** **Environment**	
	**OR (IC 95%)**	***p*-Value**	**OR (IC 95%)**	***p*-Value**	**OR (IC 95%)**	***p*-Value**
Locomotion	0.95 (0.27–3.34)	0.931	2.70 (0.82–8.91)	0.103	1.29 (0.40–4.14)	<0.001
Positions	3.42 (0.87–13.42)	0.078	1.61 (0.48–5.37)	0.436	2.16 (0.64–7.29)	<0.001
Balance	16.41 (4.06–66.35)	<0.001	7.52 (2.13–26.54)	0.002	7.88 (2.24–27.64)	<0.001
Leg Coord.	1.13 (0.29–4.42)	0.861	1.57 (0.46–5.34)	0.473	1.73 (0.51–5.87)	<0.001
Arm Coord.	1.45 (0.42–5.00)	0.558	0.76 (0.24–2.39)	0.642	1.76 (0.56–5.55)	<0.001
Hand Coord.	1.39 (0.37–5.26)	0.63	1.35 (0.38–4.76)	0.645	1.94 (0.56–6.75)	<0.001
BSA	2.77 (0.63–12.21)	0.179	2.99 (0.74–12.11)	0.126	1.70 (0.44–6.58)	<0.001
BSoA	4.76 (1.10–20.55)	0.037	2.28 (0.61–8.53)	0.219	2.77 (0.74–10.29)	<0.001
R^2^ of Nagelkerke	0.498		0.379		0.382	
Model	*χ*2(8) = 37.72; *p* < 0.001		*χ*2(8) = 27.12; *p* = 0.001		*χ*2(8) = 27.55; *p* = 0.001	

## Data Availability

Data sharing not applicable.

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
