# Peer review of "Influence of the Psychomotor Profile in the Improvement of Learning in Early Childhood Education"

_ijerph, 2021, doi:10.3390/ijerph182312655_

Round 1
Reviewer 1 Report
General comments
This study aimed to discover the influence of psychomotor profiles on academic performance throughout three to five years. Although you have already written this manuscript, my major concerns over the study and/or the manuscript include (a) introduction and its linkage with methods; (b) reliability and validity of the instruments; (c) ethical issue; (d) the presentation of results; (e) disconnection between introduction and discussion; (f) languages, references, in-text citations and formatting
Specific comments:
Line 35-39, 44-46: Could you please provide a background of how the educational curriculum is lack competent content and psychomotor skills worldwide and in your country? This may provide how important your study is, including the instrument you used and the specific curriculum that you will promote in the pre-school context. Please also specify which journal you are referred to for each sentence, or are all those sentences from the same article?
Introduction: This part is required to have a substantial re-construction. It could provide the readers to know the theoretical alignment of competence, psychomotor skills and the relationship with academic development.
Line 44-56: Would you please separate the in-text citations? Were all four (numerous) articles explaining the same thing?
Line 57-64: Your references were published in 2008 and 2015. Would you please provide some more updated publications in recent years? There were many articles relating to psychomotor skills published in 2020 and 2021. To strengthen your rationale, please determine a very specific research gap by concluding the methodology and results from previous literature.
Line 65-75: Please re-structure this paragraph to introduce the five aspects of the psychomotor profile.
Line 81-87: Is there any relationship between both statements? This paragraph is hard to follow, is it explaining the active play during recess or the regular PA or structured PE is associated with academic development or specifically, Mathematics?
Line 92-93: physical education, no capital letters are needed.
Line 99-112: Please do not provide point forms. You may explain the research gap you discovered and explain your aims cordially.
Materials and Methods: The authors should present the results in a more organized way, so you can discuss them easier in the latter paragraph.
Line 121: Please explain why your participants were selected conveniently. Where are they from, in the same pre-school?
Line 121-126: These sentences are quite hard to follow. Please separate them to indicate each issue. What do you mean by 5 lines or groups?
Line 124: What are the inclusion-exclusion criteria?
Line 128-129: Why they suspended participation in this study?
Line 131-139: Are they validated tools to assess psychomotor skills and academic performance? What language of these tools is using? Could you please provide related construction and validation results and references (reliability and validity)? This could robustly affect your study results and discussion. Please explain them in detail.
Line 141: A systematic review could only be conducted through a series of standardized procedures that your study did not indicate any one of the procedures.
Line 149: What is the measurement criteria? Is there a standardized protocol for your data collection (with reference)?
Line 158-159: What do you mean by “were always collected”?
Line 167-168, 170: Why evolution? Do you mean students psychomotor skills can only be improved? Is there any example that students’ performance was decreased?
Line 178-179: Although you specify the ethical aspects of this study, I doubt the issue of missing ethical approval.
Results:
P.4-7: Three separate tables are indicating the psychomotor skills scales, you may reconsider the methods you have used (Two-way ANOVAs) or simplify them as one table. Alternatively, you may use a table to show the descriptive data and explain the significant difference item in words. Additionally, were there any significant differences between years? There is no such meaning in discussing the difference within a year in a longitudinal study.
Line 214-223 and Table 4: You did not explain the determination of good, normal and low in the methods. The reader is quite hard to follow.
Line 225-235 and Table 5: Similar issue of Table 4.
Discussion: The authors should provide the overall picture of this study and discuss the results concerning your aims. The limitation should also be addressed.
Author Response
Thanking the contribution made, the manuscrypt has been revised and submitted.

Reviewer 2 Report
None.
Author Response
Thanking the contribution made, the manuscript has been revised and submitted.
Point 1: Reviewer has concerns regarding scientific quality in a research endeavor, since it appears to be a collection of well-know data about children psychomotor development. The promise the authors made up with the Title and Resume (offering some discoveries between psychomotor profile (sic) and cognitive features it is NOT supported by: Introduction of cognitive theoretical frameworks
Response 1: Thank you for the contribution. We have proceeded to restructure some aspects reported in the introduction in order to align the common thread in a more favourable and understandable way for the reader. Thus, an attempt has been made to further qualify the importance of the relationship between the practice of regular physical activity and structured physical education with the development of cognitive skills associated with academic performance, among which the student's concept of himself, his self-esteem, motivation and social relationships. These elements, on the other hand, are put in value when developing motor skills, since they allow the child to relate to the outside world. From here, it is easy to understand that other authors conclude that this practice of physical activity at an early age is related to academic performance in more specific subjects specifically included in the teaching curriculum such as mathematics and reading.
Point 2: Using jus raw academic performance data for to compare with psychomotor levels (actually, this kind of raw data are NOT reliable, must be used in a relative –children socio-economic background, e.g.) way.
Response 2: Thank you for the contribution. The academic results were those that the teaching staff officially registered in the corresponding report cards at the end of each quarter, following the standards established by the specific regulations on the subject and as established in the teaching program approved by the corresponding administrative entity. Having as one of the objectives of the study to know how basic competences evolve, within the same academic year and throughout the three years, through both the analysis of motor competencies and academic performance, forced to establish this relationship between variables.
Point 3: Authors seem to use "intervention" (a very specific concept related with problems to be solved) alternatively with "education" in several places of the manuscript.
Response 3: Thank you for the contribution. The authors refer to the word intervention throughout the text on two occasions:
- On line 61, stating statements by other authors regarding the importance of carrying out adequate psychomotor intervention programs from an early age to the teaching program to help complete a good comprehensive education of the child.
- On line 366, added after review, referring to the interest that the design and implementation of this intervention program could have meant for the present work, to analyse whether or not the results obtained would vary, proposing this idea as a future line of action.
Point 4: Finally, there are no University of Murcia Ethical Committee endorsing code, but just an explanation which don't assure nothing about the use of these critical children data, and its final conservation responsible.
Response 4: It should be pointed out that it is not that ethical approval is lacking, but rather that the Ethics Committee determined that as it was an observational study and the tests given to the students, it was not necessary to issue a report from this committee.
Reviewer 3 Report
GENERAL COMMENTS
Interesting longitudinal research. These types of studies have an added value and that can justify the low N. Congratulations to the authors for this 3-year follow-up.
Can you provide some more descriptive information about the characteristics of children who took the test? Locality or location, weekly physical education sessions and any data that the authors consider relevant to understand this evolution and its sociocultural context.
Reference 12 and 18 is the same. Please unify the reference and modify the bibliography accordingly.
Check the decimal signs. Normally in English it is put. and no, 0.05 and not 0,05
COMENTARIOS ESPECÍFICOS
Line 114. Authors should specify the initial N and missing data in a flow chart (figure), indicating how many started the study and how many were missing per academic year.
Line 141. This sentence is outside the scope of the current paper. “Firstly, a systematic review of the literature was carried out. Different databases were 141 consulted with the aim of developing a first approximation to the theoretical framework 142 that would serve as a basis for carrying out the fieldwork. The databases consulted in- 143 cluded Web of Science, PubMed, Science Direct, Scopus and Ebsco”. If you have done this, you should indicate in a PRISMA model figure the results of your review, if it is published, please add the reference, and if it is none, omit this paragraph.
Author Response
Thanking the contribution made, the manuscript has been revised and submitted.

Round 2
Reviewer 1 Report
Thank you for your revisions. This paper should have made more substantial revisions, especially the introduction, methods, results and discussion section, and ethical issues that may exist in this study's methodology.
Specifically, the reader may not recognise your needs of conducting this research relating to your country/ context and the rationale of this study and using those instruments, etc. The connections between each section are lacking. As such, the content and language of this paper should also edit extensively.
Author Response
Dear reviewer
We have received notification of review of our manuscript submitted to the journal.
The truth is that we do not understand that for reviewers 2 and 3, the article is already ok to be published and yet for you, it is still not.
The questions he raises are totally generic and he does not specify any specific modification, after having answered each and every one of the considerations he sent us in the first review report.
We consider that this revision is not adapted to the reality of the manuscript, for example, when he blames us for problems in the writing that are totally uncertain, such as the level of English, for example. The manuscript has been written by a native speaker and revised by the translation company TRANSPERFECT (https://www.transperfect.com/es/home), which is why we consider the rating you apply to us regarding the use of the language to be incoherent.
We therefore ask you to consider accepting the manuscript in its current state and publishing it.
Thank you very much for your attention